# Predictive Impact of Hematological and Biochemical Parameters on the Clinical Course of Sarcoidosis

**DOI:** 10.3390/diagnostics15192501

**Published:** 2025-10-01

**Authors:** Tugba Onyilmaz, Serap Argun Baris, Huseyin Kaya, Ayse Zeynep Pehlivan, Hanife Albayrak, Sena Nur Aktoprak, Hasim Boyaci, Ilknur Basyigit

**Affiliations:** 1Department of Chest Diseases, University of Kocaeli, 41001 Kocaeli, Turkey; 2Department of Chest Diseases, Kocaeli City Hospital, 41060 Kocaeli, Turkey

**Keywords:** sarcoidosis, prognosis, LAR, NLR, PLR, LMR

## Abstract

**Background:** Sarcoidosis is a systemic granulomatous disease with a highly variable clinical course, and distinguishing it from other diseases and predicting its prognosis can be challenging. In recent years, hematological and biochemical parameters have been investigated as potential biomarkers for assessing inflammation and predicting disease prognosis. This study aimed to evaluate the prognostic value of the lactate dehydrogenase-to-albumin ratio (LAR), neutrophil-to-lymphocyte ratio (NLR), platelet-to-lymphocyte ratio (PLR) and lymphocyte-to-monocyte ratio (LMR). **Methods:** This retrospective, single-center study included 369 newly diagnosed patients with sarcoidosis who were admitted between January 2020 and October 2024. Sarcoidosis was diagnosed based on current ERS, ATS, and WASOG guidelines. At the 6-month follow-up, patients’ clinical courses were classified as regression, stable, or progression based on clinical, radiological, and pulmonary function tests. The predictive values of various hematological and biochemical parameters were analyzed using statistical methods, including binary logistic regression analysis and ROC analysis. **Results:** A total of 369 patients were included in the study. At the 6-month follow-up, 63.4% of patients showed regression, 21.4% had a stable course, and 15.2% showed progression. The progression group had a significantly higher LAR (5.26 [4.21–7.76]) compared to the stable/regression group (4.59 [3.82–5.86]) (*p* = 0.033). Additionally, baseline FVC% (OR, 0.97; *p* = 0.036) and the presence of dyspnea (OR, 3.08; *p* = 0.03) were identified as independent risk factors for disease progression. No significant associations were found between NLR, PLR, LMR, and serum ACE levels and the clinical course. The cutoff value of LAR for predicting disease progression was 4.87 (AUC: 0.605), with a sensitivity of 58.8% and specificity of 59.7%. **Conclusions:** Our study, which is the first to evaluate the prognostic value of LAR in sarcoidosis, identified this parameter as a significant indicator for the clinical course. The finding of significantly higher LAR levels in patients with disease progression suggests its potential as a prognostic biomarker. These results may help guide treatment and follow-up strategies, although further large-scale prospective studies are needed for validation.

## 1. Introduction

Sarcoidosis is a systemic granulomatous disease characterized by noncaseating granulomas that frequently involve the respiratory system. Although the exact etiology remains unknown, it is a multisystem condition that can affect various organs. The clinical course of sarcoidosis can vary widely; the disease may cause no symptoms in some individuals, while in others it can lead to serious multiple organ involvement. In approximately 30–50% of cases, patients show no symptoms, and the disease is usually detected incidentally on chest X-rays [1].

Different epigenetic mechanisms may play a role in the onset, prevalence, and course of sarcoidosis. Current research shows that environmental, occupational, infectious, and socioeconomic factors combine to contribute to the development of the disease in genetically predisposed individuals [2].

Sarcoidosis is diagnosed based on a combination of clinical, radiological, laboratory, and histopathological findings [3]. However, differentiating it from other granulomatous diseases and predicting its prognosis can be challenging, including tuberculosis and fungal infections that present with granulomas and sarcoid-like reactions, lymphomas, various malignancies, such as lung and breast cancer, drug-induced reactions, and immunodeficiency syndromes [4].

Inflammation plays a critical role in the pathogenesis and progression of sarcoidosis. Although some patients experience spontaneous regression, others may develop life-threatening organ involvement. After a diagnosis is established, determining which patients require treatment can be challenging [5].

Various biomarkers, such as serum angiotensin-converting enzyme (sACE), interleukin-2, chitotriosidase, neopterin, and YKL-40, have been investigated to overcome the challenges encountered in diagnosing sarcoidosis, making treatment decisions, and predicting disease progression [6]. Literature reviews indicate that although these biomarkers are not considered gold standards individually, they may be potentially useful in the diagnosis and monitoring of sarcoidosis due to their considerable sensitivity and specificity. Among these, sACE is the first and most commonly used biomarker in routine clinical practice [7].

Angiotensin-converting enzyme (ACE) converts angiotensin I to angiotensin II, thereby helping to maintain blood pressure and electrolyte balance. This enzyme is produced by epithelial cells and macrophages and contributes to granuloma formation in sarcoidosis. Therefore, sACE levels are considered a biomarker reflecting granuloma burden. However, its diagnostic sensitivity and specificity vary, ranging from 29% to 90.5% and from 47% to 89.9%, respectively. Elevated sACE levels can also be observed in other granulomatous diseases, such as tuberculosis, leprosy, silicosis, and primary biliary cirrhosis, as well as in some non-granulomatous conditions and malignancies [4]. In patients with active sarcoidosis, sACE levels are significantly higher than in those in remission or in healthy individuals. Some studies have demonstrated that sACE reflects disease activity, with a sensitivity of 88% and specificity of 47%. The highest levels are observed in radiological stages 1–2, and sACE has been reported to correlate with changes in pulmonary function tests, such as DLCO and serum soluble IL-2 receptor levels. Additionally, ACE levels in bronchoalveolar lavage (BAL) fluid may also increase and are suggested to reflect pulmonary disease activity more accurately than serum ACE levels, particularly in stage 2–3 sarcoidosis [8].

Neutrophils, lymphocytes and platelets are key cellular components of the inflammatory response. Neutrophils serve as the first line of defense in the early phase of inflammation, whereas lymphocytes play a role in regulating inflammation and sustaining the immune response. Owing to their ability to secrete chemokines and cytokines, platelets also play significant roles in inflammation, and the neutrophil-to-lymphocyte ratio (NLR) has been reported to increase in many inflammatory diseases. In addition, the platelet-to-lymphocyte ratio (PLR) and lymphocyte-to-monocyte ratio (LMR) are considered biomarkers of systemic inflammation. In patients with sarcoidosis, these parameters have been shown to be useful in disease staging and in identifying poor prognosis [9,10]. Several studies have identified the lactate dehydrogenase to albumin ratio (LAR), which allows for the simultaneous assessment of LDH and albumin levels as an economical, accessible, and reproducible parameter that can serve as a valuable tool for prognostic evaluation in lower respiratory tract infections, COVID-19, pulmonary malignancies, and respiratory diseases, such as pulmonary embolism [11,12].

This study aimed to evaluate hematological and biochemical parameters in patients with sarcoidosis and to assess the predictive value of LAR, NLR, PLR and LMRs in determining disease progression and remission in these patients.

## 2. Materials and Methods

### 2.1. Study Population

This retrospective study includes patients who were newly diagnosed with sarcoidosis in the Pulmonary Diseases Department between January 2020 and October 2024. Sarcoidosis was diagnosed based on a combination of clinical, radiological, laboratory, and/or histopathological findings in accordance with the current ERS, ATS, and WASOG guidelines [5,13].

Radiological staging was performed based on chest radiographs obtained at the time of diagnosis using the scadding staging system. This classification consists of five stages, ranging from bilateral hilar lymphadenopathy (stage 1) to fibrotic changes (stage 4) and is widely used to indicate the extent of pulmonary involvement in sarcoidosis [13].

Exclusion criteria included other respiratory diseases, such as asthma, lung cancer, bronchiectasis, interstitial lung diseases, and corticosteroid use. In addition, patients who had experienced an infection and/or had received antibiotic therapy within the past month were excluded from the study to avoid potential effects on laboratory parameters.

Patients requiring treatment were initiated on oral methylprednisolone at a starting dose of 32 mg/day in accordance with the standard protocol. During follow-up, treatment was continued with gradual dose tapering according to clinical guidelines. Among the patients who received treatment and showed disease progression, it was determined that they were on a maintenance dose of methylprednisolone ranging between 4 and 8 mg/day [14].

### 2.2. Data Collection

Patients’ demographic data, initial radiological stages, physical examination findings, comorbidities, smoking history, routine laboratory parameters, spirometry values, carbon monoxide diffusion test (DLCO) results, bronchoalveolar lavage (BAL) lymphocyte percentages, and lymphocyte subset analyses were recorded. The evaluated laboratory variables included acute-phase reactants (CRP and erythrocyte sedimentation rate), complete blood count, blood glucose, renal function tests (creatinine and urea), liver function tests (AST and ALT), alkaline phosphatase (ALP), serum calcium, 24 h urinary calcium, and ACE levels. In addition, to assess the clinical course of the disease, patients’ status at 6 months after diagnosis was classified as stable, progression, or regression based on clinical, radiological, pulmonary function tests, and DLCO values. Radiological progression was defined as progression of sarcoidosis stage, or, in patients with Stage 3 and Stage 4 disease, as an increase of more than 10% in parenchymal findings with or without worsening clinicals symptoms. Pulmonary function test progression was defined as a decline of ≥20% in FVC and/or ≥10% in DLCO with or without worsening clinicals symptoms. Symptom worsening alone was not considered progression.

This study has been ethically approved by the Ethics Committee of Kocaeli University in accordance with the principles of the Helsinki Declaration (Ethics Committee decision: KU GOKAEK-2024/20/13; Project code: 2024/488; Approval date: 24 December 2024). Since the study was planned retrospectively, written informed consent was not required.

### 2.3. Statistical Analysis

Statistical analyses were performed using IBM SPSS version 25.0 (SPSS Corp., Armonk, NY, USA). The Kolmogorov–Smirnov test was used to assess the normality of the distribution. Continuous covariates was reported as Median (25th–75th percentiles). Categorical variables are expressed as frequencies (percentages). For comparisons between two independent groups, the Mann–Whitney U test was used, whereas the Kruskal–Wallis test was applied for comparisons involving more than two groups. Associations between categorical variables were evaluated using Pearson’s chi-square test. Binary logistic regression analysis was performed to identify the predictors of disease progression, and odds ratios (ORs) with 95% confidence intervals (CIs) were calculated. The diagnostic performance of LAR was evaluated using a receiver operating characteristic (ROC) curve, along with positive and negative predictive values. Statistical significance was set at *p* < 0.05.

## 3. Results

A total of 369 patients (268 female [72.6%], 101 male [27.4%]) with a median age of 52 (43–61) years were included in the study. Radiological staging of the patients was as follows: stage 0, 13 (3.5%); stage 1, 101 (27.4%); stage 2, 198 (53.7%); stage 3, 18 (4.9%); and stage 4, 39 (10.6%). Extrapulmonary involvement was present in 39.5% of patients, with the most frequently affected sites being the skin (18.8%) and peripheral lymph nodes (13.4%).

A total of 205 patients (55.6%) received treatment, while 164 (44.4%) were followed up without treatment. Seventy-four percent of patients were asymptomatic. The most common symptoms were dyspnea (39%) and cough (36.5%). At the time of admission, 21.2% of patients were current smokers and 8.5% had a family history of pulmonary disease. On physical examination, 82.2% of patients had no pathological findings, while crackles were present in 10.6% and rhonchi in 7.2% of patients.

Pulmonary function tests revealed normal results in 194 patients (79.2%), obstructive patterns in 28 (11.4%), restrictive patterns in 14 (5.7%), and mixed-type disorder in 9 (3.7%).

At the 6-month follow-up, radiological regression was observed in 234 patients (63.4%), disease remained stable in 79 patients (21.4%), and progression occurred in 56 patients (15.2%) (Table 1). Among those with progression, 11 patients (20%) had radiological progression alone, whereas 45 patients (80%) exhibited significant deterioration in both radiological findings and pulmonary function tests. Worsening respiratory symptoms were noted in patients with deterioration in both radiological and functional parameters.

Patients were grouped according to their clinical course into two categories: progression and stable/regression groups. The median age of patients in the progression group 56 (42.5-65) was higher than that of the stable/regression group 52 (43-60.5), although this difference was not statistically significant (*p* = 0.083).

Among the patients with stable or regressive disease, 85.7% were in stage 1 or 2, whereas stage 4 patients constituted 39.3% of the progression group, which was significantly higher than the 5.4% in the stable/regression group (*p* < 0.000) (Table 2). Progression was observed to be more prominent particularly in Stage 3 and Stage 4 patients. Notably, the percentage of progressive disease in Stage 4 was higher compared to the other groups, and this difference was found to be statistically significant (Figure 1).

There was no statistically significant correlation between baseline serum ACE levels and clinical course at the 6th month (*p* = 0.161). The proportion of patients receiving treatment in the progression group (78.6%) was significantly higher than that in the stable/regression group (51.4%) (*p* = 0.000). Additionally, respiratory symptoms (cough and dyspnea) (*p* < 0.000) and abnormal physical findings (crackles and rhonchi) (*p* < 0.001) were observed more frequently in the progression group.

In terms of pulmonary function, the baseline FEV1%, FVC%, DLCO, and DLCO% values were significantly lower in the progression group than in the stable/regression group (*p* < 0.001, *p* = 0.001, *p* = 0.020, and *p* = 0.046, respectively).

Among the laboratory parameters, neutrophil count was significantly higher in the progression group (5100 [3.7–6.9]) than in the stable/regression group (4206 [3.2–5.5]) (*p* = 0.021). Additionally, the LAR was significantly elevated in the progression group (5.26 [4.21–7.76]) compared to the stable/regression group (4.59 [3.82–5.86]) (*p* = 0.033) (Table 2).

This study also evaluated the predictive potential of inflammation and immune response-related biomarkers NLR, PLR, and LMR for disease progression in sarcoidosis. However, there were no statistically significant differences between the progression and stable/regression groups for these markers (NLR, *p* = 0.105; PLR, *p* = 0.368; and LMR, *p* = 0.435).

In the binary logistic regression analysis, baseline FVC% (OR, 0.97; 95% CI, 0.937–0.998; *p* = 0.036) and presence of dyspnea (OR, 3.08; 95% CI, 1.117–8.48; *p* = 0.03) were identified as independent risk factors for disease progression (Table 3).

According to ROC analysis, the cutoff value of LAR for predicting disease progression was determined to be 4.87 (AUC: 0.605; 95% CI: 0.508–0.703; *p* = 0.033), with a sensitivity of 58.8% and a specificity of 59.7% at this threshold (Figure 2).

## 4. Discussion

In this study, both the LAR value and neutrophil count were found to be significantly higher in the group that showed progression at the 6-month follow-up. In the same group, the baseline FEV_1_%, FVC%, DLCO, and DLCO% values were lower than those in the regression+ stable group. However, no statistically significant association was observed between baseline serum ACE levels and clinical course. This study contributes to the literature by evaluating the prognostic value of the LAR in sarcoidosis.

An ideal biomarker for assessing disease activity should be simple, low-cost, easy to apply, reproducible, and repeatable and should accurately reflect the current level of disease activity with high sensitivity and specificity [10]. In this context, we aimed to evaluate the predictive role of LAR and other readily accessible hematologic and biochemical parameters, such as NLR, PLR and LMR in the clinical course of sarcoidosis.

The significantly higher LAR values observed in sarcoidosis patients who showed disease progression at the 6-month follow-up suggest that this parameter may reflect systemic inflammation. LDH is an enzyme that is widely found in human cells and catalyzes the conversion of lactic acid to pyruvic acid. It increases in the context of cellular damage and acute inflammatory processes and may increase as a result of tissue injury associated with increased granuloma burden [15,16,17]. Albumin, in addition to maintaining plasma oncotic pressure, is a protein with anti-inflammatory properties, and its serum level tends to decrease during inflammatory processes and states of high energy consumption [18]. Hypoalbuminemia is associated with poor prognosis, prolonged hospitalization, and increased mortality risk [19,20]. Since both LDH and albumin can be influenced by various factors, such as liver disease, proteinuria, age, and nutritional status, their sensitivity and specificity are limited when used alone. However, a combination of these two parameters in the form of LAR may offer more reliable prognostic information. Although there is strong evidence supporting the prognostic value of LAR in malignancy, COVID-19, sepsis, and other respiratory diseases, data regarding its role in sarcoidosis are currently lacking [20,21,22]. According to the ROC analysis, although the sensitivity and specificity values of LAR in predicting disease progression were not high, the LAR value was found to be significantly elevated in the group with progression in our study.

When the findings of our study are evaluated in terms of their clinical implications, it appears that patients with high LAR values carry a higher risk of disease progression. Therefore, even if other parameters suggest a watch-and-wait approach, the LAR may help clinicians in deciding whether to initiate treatment or not, supporting a more individualized management strategy. However, further large-scale prospective studies are needed to better define the role of LAR in routine clinical decision-making. Given that sarcoidosis is a multifaceted disease, it cannot be assessed based on a single parameter alone. Nonetheless, we believe that the LAR could serve as an additional guiding parameter for clinicians when evaluating treatment decisions.

In recent years, various studies have investigated NLR in sarcoidosis patients [23,24]. In a meta-analysis conducted by Alamdari et al., the NLR was suggested as a potential prognostic marker for assessing the radiological severity of pulmonary sarcoidosis, aiding in the evaluation of lung parenchymal involvement, and even in diagnosing pulmonary hypertension [25]. In our study, however, no statistically significant relationship was found between baseline NLR and disease progression at the 6-month follow-up. This discrepancy may be attributed to methodological differences; while the studies included in the meta-analysis primarily examined the relationship between baseline NLR and extrapulmonary involvement, our study focused on the 6-month clinical course. Tartemiz et al. reported that high NLR, high PET/CT SUVmax, and low DLCO% values were predictive of disease progression over a one-year follow-up period. In contrast, our study did not demonstrate a statistically significant association between the baseline NLR and progression, regression, or stability at 6 months. This inconsistency may be due to the shorter follow-up period (6 months instead of 12 months) and the differences in treatment exposure and therapeutic regimens within our study population [26].

Platelet counts increase in response to inflammation and tissue injury, and this elevation has been shown to positively correlate with pro-inflammatory cytokines, such as TNF-α and IL-6. PLR is a simple and practical biomarker that reflects both coagulation and immune response processes simultaneously and is considered a valuable tool for predicting adverse clinical outcomes, as it reflects the interaction between inflammation, coagulation, and cytokine production [27,28]. Accordingly, we aimed to evaluate the prognostic value of PLR in predicting the 6-month clinical course of patients with sarcoidosis. However, similar to the findings reported by Korkmaz et al., no statistically significant association was observed between PLR and disease progression in our study [9].

The LMR is a relatively novel biomarker that reflects systemic inflammation. In recent years, its association with disease severity and prognosis has been investigated for the assessment of inflammatory activity, particularly in autoimmune and other granulomatous diseases [29]. In a study conducted by Masoud et al., LMR levels were higher in patients with active sarcoidosis than in those with inactive disease [10]. However, similar to our findings, Ozdemirel et al. found no significant association between the LMR and disease progression [30].

In our study, no significant association was observed between serum ACE levels and disease progression. ACE is an enzyme secreted by active granulomas and is frequently used for the diagnosis and follow-up of sarcoidosis. However, the ability of this biomarker to reflect disease activity and progression is limited. While some studies have demonstrated a correlation between ACE levels and disease activity or extent in sarcoidosis, others have failed to confirm this relationship. It has been shown that in fibrotic stages, ACE levels may remain within normal ranges owing to reduced granuloma burden. Additionally, factors such as genetic polymorphisms and the use of medications, such as ACE inhibitors, may also affect serum ACE levels [8]. Therefore, ACE alone is not a reliable biomarker for predicting sarcoidosis progression.

Our study has several limitations. First, the single-center and retrospective design limits the generalizability of the findings. In addition, the lack of evaluation of other potential biomarkers that may be associated with prognosis, such as chitotriosidase and IL-2R, represents another important limitation. Since the clinical course was assessed only at the 6-month follow-up, no information could be obtained regarding long-term prognosis or temporal changes in laboratory parameters. Furthermore, due to the retrospective design and missing data at the 6-month follow-up, it was not possible to evaluate isolated progression types (clinical, radiological, and pulmonary function test) or the relationship between baseline LAR values and 6-month LAR levels in treated patients. Nevertheless, one of the strengths of our study is the inclusion of a large number of patients, which increases the reliability and interpretability of our findings.

## 5. Conclusions

In our study, the LAR value was a significant parameter associated with the prognosis of patients with sarcoidosis. The observation of significantly higher LAR levels in patients with disease progression suggests that this parameter could serve as a prognostic biomarker in clinical practice. Furthermore, patients with advanced-stage disease, reduced baseline pulmonary function, and respiratory symptoms were found to have a higher risk of disease progression. These findings may play an important role in guiding treatment and follow-up strategies for sarcoidosis. However, further large-scale prospective studies are needed to validate these results and facilitate their translation into clinical practice.

## Figures and Tables

**Figure 1 diagnostics-15-02501-f001:**
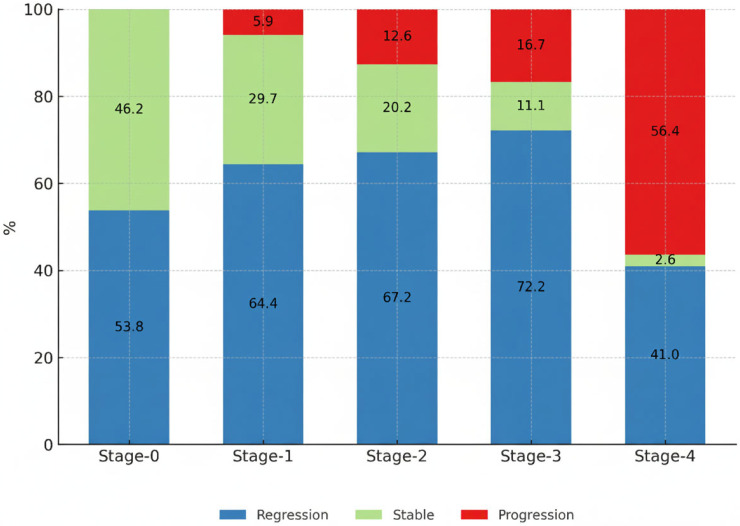
Clinical course at the 6th month according to disease stage.

**Figure 2 diagnostics-15-02501-f002:**
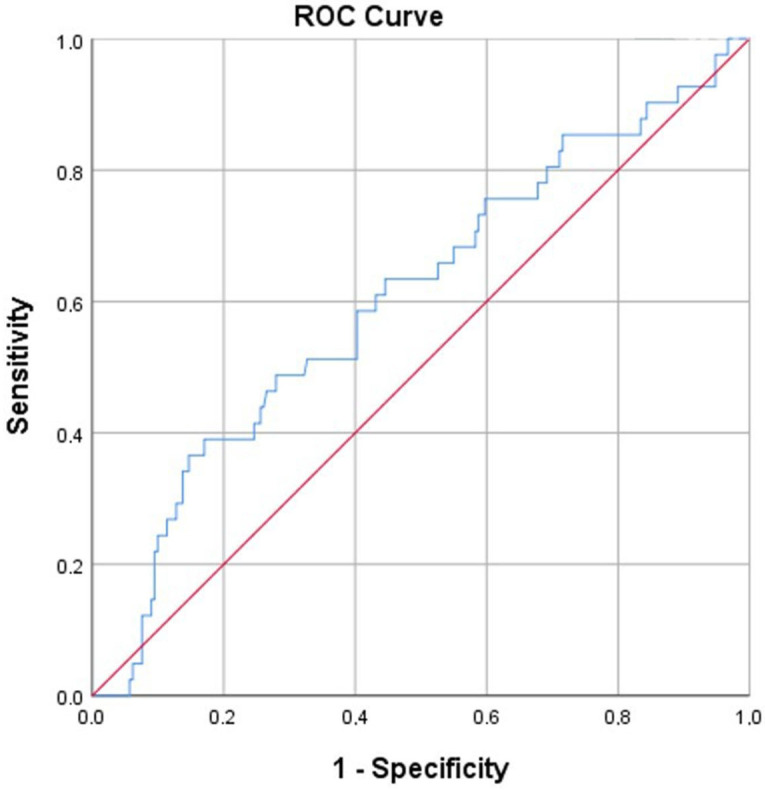
Receiver operating characteristic (ROC) curve analysis of lactate dehydrogenase-to-albumin ratio for predicting disease progression in patients with sarcoidosis at 6-month follow-up. The optimal cutoff value was 4.87 (AUC = 0.605, sensitivity = 58.8%, specificity = 59.7%).

**Table 1 diagnostics-15-02501-t001:** Demographic characteristics of the study population.

		N (%)
Gender	Female	268 (72.6)
	Male	101 (27.4)
Age, years, Median/(25th–75th percentiles)	52 (43–61)	
Stage	0	13 (3.5)
	1	101 (27.4)
	2	198 (53.7)
	3	18 (4.9)
	4	39 (10.6)
Treatment	Yes	205 (55.6)
	No	164 (44.4)
Clinical course of disease at 6th month	Regression	234 (63.4)
	Stable	79 (21.4)
	Progression	56 (15.2)
Symptoms	Asymptomatic	273 (74)
	Weight loss	19 (5.2)
	Fatigue	82 (22.6)
	Loss of appetite	4 (1.1)
	Fever	3 (0.8)
	Respiratory Symptoms	223 (60.4)
	Cough	133 (36.5)
	Dyspnea	142 (39)
	Chest pain	50 (13.7)
	Hemoptysis	2 (0.5)
Smoking history	Nonsmoker	166 (61.7)
	Current Smoker	57 (21.2)
	Former Smoker	46 (17.1)
Family History	Yes	17 (8.5)
	No	182 (91.5)
Physical examination	Normal	295 (82.2)
	Crackles	38 (10.6)
	Rhonchi	26 (7.2)
Respiratory function test	Normal	194 (79.2)
	Obstructive	28 (11.4)
	Restrictive	14 (5.7)
	Mixed	9 (3.7)
Granuloma with FOB	Yes	176 (74.6)
	No	60 (25.4)
Extra pulmonary involvement	+	145 (39.5)
	Peripheral LAP	49 (13.4)
	Skin	69 (18.8)
	Eye	21 (5.8)
	Cardiovascular	3 (0.8)
	Neurological	2 (0.5)
	Musculoskeletal	19 (5.2)
	Gastro-urinary	26 (7.1)
	Hematopoietic system	5 (1.4)

Abbreviations: FOB: Fiberoptic bronchoscopy; LAP: Lymphadenopathy.

**Table 2 diagnostics-15-02501-t002:** Clinical course of disease at 6th month.

		Regression + Stable N:313	Progression N:56	*p*
Demographic characteristics				
Gender	Female	230 (73.5%)	38 (67.9%)	0.39
	Male	83 (26.5%)	18 (32.1%)	
Age, years	Median/(25th–75th percentiles)	52 (43–60.5)	56 (42.5–65)	0.053
Stage	0	13 (4.2%)	0 (0%)	0.000
	1	95 (30.4%)	6 (10.7%)	
	2	173 (55.3%)	25 (44.6%)	
	3	15 (4.8%)	3 (5.4%)	
	4	17 (5.4%)	22 (39.3%)	
Treatment	No	152 (48.6%)	12 (21.4%)	0.000
	Yes	161 (51.4%)	44 (78.6%)	
Symptoms	Asymptomatic	241 (77%)	32 (57.1%)	0.002
	Weight loss	17 (5.5%)	2 (3.7%)	0.58
	Fatigue	59 (19.1%)	23 (42.6%)	0.000
	Loss of appetite	4 (1.3%)	0	0.4
	Fever	3 (1%)	0	0.47
	Respiratory Symptoms	176 (56.2%)	47 (83.9%)	0.000
	Cough	102 (32.9%)	31 (57.4%)	0.001
	Dyspnea	106 (34.2%)	36 (66.7%)	0.001
	Chest pain	41 (13.2%)	9 (16.7%)	0.49
	Hemoptysis	1 (0.3%)	1 (1.9%)	0.16
Physical examination	Normal	266 (86.6%)	29 (55.8%)	0.000
	Crackles	20 (6.5%)	18 (34.6%)	
	Rhonchi	21 (6.8%)	5 (9.6%)	
Extra pulmonary involvement	+	128 (41%)	17 (30.9%)	0.16
	Peripheral LAP	43 (13.8%)	6 (10.9%)	0.56
	Skin	62 (19.9%)	7 (12.7%)	0.21
	Eye	19 (6.1%)	2 (3.6%)	0.46
	Cardiovascular	3 (1%)	0	0.47
	Neurological	0	0	NA
	Musculoskeletal	18 (5.8%)	1 (1.8%)	0.22
	Gastro-urinary	23 (7.4%)	3 (5.5%)	0.61
	Hematopoietic	4 (1.3%)	1 (1.8%)	0.7
Laboratory tests at the time of diagnosis				
Creatinine (mg/dL)	Median/(25th–75th percentiles)	0.7 (0.6–0.8)	0.7 (0.6–0.8)	0.176
Calcium (mg/dL)	Median/(25th–75th percentiles)	9.5 (9.1–9.8)	9.6 (9.4–9.8)	0.303
24 h urinary calcium (mg/day)	Median/(25th–75th percentiles)	185 (106.5–265)	168 (70–321)	0.534
Albumin (g/L)	Median/(25th–75th percentiles)	42 (39–44.5)	41.6 (37.5–44)	0.138
ACE (U/L)	Median/(25th–75th percentiles)	71 (41–103)	55 (30–81)	0.161
Neutrophil count, ×10^3^	Median/(25th–75th percentiles)	4.206 (3.2–5.5)	5.100 (3.7–6.9)	0.021
LDH (U/L)	Median/(25th–75th percentiles)	189 (162–220)	218 (179–272)	0.058
NLR	Median/(25th–75th percentiles)	2.64 (1.9–3.5)	3.09 (1.9–5)	0.105
PLR	Median/(25th–75th percentiles)	172.1 (125.9–219.04)	150.5 (120.7–202.5)	0.368
LMR	Median/(25th–75th percentiles)	2.71 (2.08–3.75)	2.69 (1.84–3.49)	0.35
LAR	Median/(25th–75th percentiles)	4.59 (3.82–5.86)	5.26 (4.21–7.76)	0.033
FEV1	Median/(25th–75th percentiles)	2.43 (2.05–2.88)	2.32 (1.78–3.08)	0.59
FEV1 %	Median/(25th–75th percentiles)	92 (79–102.3)	78 (68.5–92.5)	0.001
FVC	Median/(25th–75th percentiles)	3.0 (2.5–3.6)	2.96 (2.15–3.82)	0.62
FVC %	Median/(25th–75th percentiles)	94 (84–105)	81 (73.5–94.5)	0.001
FEV1/FVC	Median/(25th–75th percentiles)	82 (77–86)	81 (75–84)	0.812
DLCO (mmol/kPa/min)	Median/(25th–75th percentiles)	6.3 (5.4–9.8)	5.1 (3.7–6.04)	0.020
DLCO %	Median/(25th–75th percentiles)	75 (66–88)	70 (62–83)	0.046

Abbreviations: LAP: lymphadenopathy; ACE: angiotensin-converting enzyme; LDH: lactate dehydrogenase; NLR: neutrophil-to-lymphocyte ratio; PLR: platelet-to-lymphocyte ratio; LMR: lymphocyte-to-monocyte ratio; LAR: lactate dehydrogenase-to-albumin ratio; FEV1: forced expiratory volume in one second; FVC: forced vital capacity; DLCO: diffusing capacity of the lung for carbon monoxide.

**Table 3 diagnostics-15-02501-t003:** In binary logistic regression analysis; risk factors associated with progression in patients with sarcoidosis.

Variable	OR [95% CI]	*p*
Treatment	1.398 [0.468–4.178]	0.549
Initial FVC %	0.967 [0.937–0.998]	0.036
Initial Neutrophil count	1.0 [1.0–1.01]	0.751
LDH/Albumin (LAR)	0.972 [0.902–1.048]	0.462
Cough	1.885 [0.754–4.713]	0.175
Dyspnea	3.078 [1.117–8.484]	0.03
Fatigue	1.433 [0.534–3.84]	0.475

Abbreviations: FVC: Forced Vital Capacity; LDH: Lactate Dehydrogenase.

## Data Availability

The original contributions presented in this study are included in the article. Further inquiries can be directed to the corresponding author.

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
