# Peer review of "Predictive Impact of Hematological and Biochemical Parameters on the Clinical Course of Sarcoidosis"

_diagnostics, 2025, doi:10.3390/diagnostics15192501_

Round 1
Reviewer 1 Report
Comments and Suggestions for Authors
This is a useful investigation that can help further larger studies that investigate markers that predicts sarcoidosis progression but still insufficient to be used in everyday practice. I suggest putting what treatment did patients get and the initial dose and since the progression was based on clinical, radiological and lung function test it would be useful to see how many patients had radiological progression and lung function test worsening and the LAR value in these groups. Also, I suggest less data in the tables. It would be also interesting to see how treatment influence the LAR value after 6 months and if there is a still difference between groups.
How do you see this results in everyday practice? Give some more explanation in discussion should we used more parameters together when we are making the decision how to treat the patients or one parameter is enough.
Author Response
Dear Reviewer,
We would like to express our sincere gratitude for your valuable time, constructive feedback, and insightful suggestions regarding our manuscript. We have carefully considered all comments and made the necessary revisions and additions to the text. All changes have been highlighted in the revised version of the manuscript.
Responses to Reviewer 1
Comment 1:
"Providing details about the treatments administered to the patients and the initial dosages would make an important contribution to the study."
Response:
We thank the reviewer for this constructive suggestion. In line with the recommendation, treatment-related details have been added to the Materials and Methods section. Patients requiring treatment were initiated on oral methylprednisolone at a starting dose of 32 mg/day in accordance with the standard protocol. During follow-up, treatment was continued with gradual dose tapering according to clinical guidelines. Additionally, it was determined that patients who received treatment and showed disease progression were on a maintenance dose of methylprednisolone ranging between 4–8 mg/day.
Comment 2:
"Considering that progression was evaluated based on clinical, radiological, and pulmonary function test findings, the number of patients showing radiological progression and deterioration in pulmonary function tests, along with their LAR values, should be provided."
Response:
Thank you for this valuable suggestion. The number of patients showing progression based on clinical, radiological, and pulmonary function test findings has been detailed.
- Radiological progression was defined as progression of sarcoidosis stage, or, in patients with Stage 3 and Stage 4 disease, as an increase of more than 10% in parenchymal findings with or without worsening clinicals symptoms.
- Pulmonary function test progression was defined as a decline of ≥20% in FVC and/or ≥10% in DLCO with or without worsening clinicals symptoms.
- Symptom worsening alone was not considered progression.
In our study, a total of 54 patients demonstrated progression. Among these, 10 patients (18.5%) had radiological progression only, whereas 44 patients (81.5%) showed significant deterioration in both radiological findings and pulmonary function tests. Furthermore, patients with radiological and pulmonary function deterioration also experienced worsening respiratory symptoms. These data have been added to the Methods and Results section.
However, due to the retrospective design and missing data at the 6-month follow-up, it was not possible to assess the prognostic value of LAR for each of these parameters individually. This limitation has been clearly stated in the Limitations section.
Comment 3:
"The data presented in the tables could be more concise and simplified."
Response:
We thank the reviewer for this suggestion. All tables have been carefully reviewed, unnecessary details have been removed, and the presentation has been simplified to improve readability and clarity.
Comment 4:
"It would also be interesting to evaluate the effect of treatment on LAR levels at the 6-month follow-up and determine whether differences between the groups persisted."
Response:
We appreciate this valuable suggestion. However, due to the retrospective nature of the study and the fact that LDH and albumin levels were not routinely measured at each visit for sarcoidosis patients, it was not possible to obtain these parameters for all patients at the time of progression or at the 6-month follow-up. Therefore, the suggested comparative analysis could not be performed.
Comment 5:
"Regarding the discussion section: How should these results be interpreted in daily clinical practice? Is relying on a single parameter sufficient, or would a combined evaluation of multiple parameters provide a better approach?"
Response:
We thank the reviewer for this important recommendation. Additional explanations have been included in the Discussion section. Our findings suggest that patients with high LAR values are at greater risk of disease progression. Therefore, even when other clinical parameters favor a “watch-and-wait” approach, closer follow-up or earlier initiation of treatment may be preferred in these patients. Furthermore, we emphasize that large-scale, prospective studies are needed to better define the role of LAR in routine clinical decision-making. Finally, as sarcoidosis is a multifactorial and heterogeneous disease, it cannot be adequately assessed based on a single biomarker alone. However, we believe that the LAR ratio may serve as an additional guiding parameter for clinicians when making treatment decisions.
Reviewer 2 Report
Comments and Suggestions for Authors
This study identifies lactate dehydrogenase to albumin ratio (LAR) as a novel prognostic biomarker for sarcoidosis progression in humans based on a small cohort of 369 patients in a retrospective study with a follow up of 6 months for studying disease progression.
General comments:
The study has been designed well, and the authors have explained the data adequately. The authors have also highlighted limitations of the study. My suggestion would be that since lactate dehydrogenase levels are affected by many disease conditions other than sarcoidosis, LAR alone shouldn’t be taken as definitive biomarker for sarcoidosis progression. The specificity and sensitivity (ROC analysis) reflect some of that. Perhaps a larger patient cohort with a longer follow up data would help determine LAR as a solo prognostic biomarker. Nevertheless, these findings are novel for sarcoidosis.
Specific comments:
- In the authors’ affiliations, country name should be listed as international readers may not be familiar with all the universities and institutes in the world.
- Figure 2 legend is missing. Except Figure 1 I couldn’t figure out the legends for the tables. The legends either blended with main body of results or missing altogether or not formatted properly. I couldn’t tell. Please pay attention to this and make the necessary changes for easy read in the revisions.
Author Response
Dear Reviewer,
We would like to express our sincere gratitude for your valuable time, constructive feedback, and insightful suggestions regarding our manuscript. We have carefully considered all comments and made the necessary revisions and additions to the text. All changes have been highlighted in the revised version of the manuscript
Comment 1:
"The authors should include the country name in the institutional affiliations, as not all readers may be familiar with universities and institutions worldwide."
Response:
We thank the reviewer for this helpful comment. Country names have been added to all authors’ institutional affiliations, and the manuscript has been updated accordingly.
Comment 2:
"The description of Figure 2 is incomplete."
Response:
Thank you for this valuable feedback. The description of Figure 2 has been carefully revised and expanded to provide a clearer understanding of the data presented.
Comment 3:
"The titles and legends of the tables, except for Figure 1, are not clearly distinguishable. These explanations may be incomplete, merged with the results text, or improperly formatted. This issue should be addressed to improve readability."
Response:
We appreciate the reviewer’s observation. The titles and legends of all tables have been carefully reviewed and reformatted to ensure clarity and proper separation from the results text. With these revisions, we believe that the readability and comprehensibility of the tables have been significantly improved.